# Virtual Leadership and Nurses’ Psychological Stress during COVID-19 in the Tertiary Hospitals of Pakistan: The Role of Emotional Intelligence

**DOI:** 10.3390/healthcare11111537

**Published:** 2023-05-25

**Authors:** Fahad Alam, Qing Yang, Aušra Rūtelionė, Muhammad Yaseen Bhutto

**Affiliations:** 1School of Economics and Management, University of Science and Technology Beijing, Beijing 100083, China; fahadalam@xs.ustb.edu.cn; 2Faculty of Bioeconomy Development, Vytautas Magnus University, LT-44248 Kaunas, Lithuania; 3Business School, Shandong Jianzhu University, Jinan 250101, China

**Keywords:** virtual leadership, emotional intelligence, work stress, work burnout, job performance

## Abstract

Although intelligence has been widely examined in the literature, the correlation of emotional intelligence (EI) has with virtual leadership, work stress, work burnout, and job performance in the nursing profession needs further consideration. Prior studies have confirmed that leadership style and emotional intelligence massively contribute to better outcomes in the nursing profession. Based on these confirmations, this research intended to explore the impact of virtual leadership and EI on work stress, work burnout, and job performance among nurses during the COVID-19 pandemic. A convenient sampling technique was adopted to select the data sample. To analyze our hypotheses, 274 self-reported surveys were distributed in five tertiary hospitals in Pakistan through a cross-sectional quantitative research design. The hypotheses were tested with SmartPLS-3.3.9. Our findings revealed that virtual leadership and EI have considerably influenced nurses’ work stress, burnout level, and job performance. The study concludes that EI significantly moderates virtual leadership and psychological stress among nurses.

## 1. Introduction

Nurses play a significant role in healthcare, as approximately 60% of healthcare workers are nurses [1]. Frequently, nurses are exposed to critical psychological factors that can make their tasks difficult. Psychological risks are directly related to health problems, work accidents [2], dissatisfaction, poor work engagement [3], work stress, burnout, and low performance [4]. Nurses experience high levels of stress when faced with work overtime, anxiety, burnout, loneliness, and confronting patients’ sentiments [5]. The COVID-19 emergency has exerted unprecedented psychological stress and various problems on the nursing workforce. The enormous increase in COVID-19-infected patients caused a severe threat to nurses’ psychological and physical well-being [6]. This had an adverse effect on nurses’ work behavior [5]. It is critical to proactively manage nurses’ job stress, reduce burnout, and improve job performance through providing appropriate interventions and support [7].

Scholars have observed the impact of workplace stress on the psychological and work behavior of employees; for instance, Wang et al. [8] highlighted that a toxic work environment creates health issues that affect the decision-making ability and job performance of employees. A study by Khattak et al. [5] revealed depression, anxiety, and insomnia among nurses operating during the COVID-19 crisis. A study by Barello et al. [9] indicated that an increased fear of COVID-19 engenders psychological stress and turnover intention and decreases job performance in the nursing profession [10]. Maintaining a nurse’s psychological well-being is important [11].

With the outbreak of the COVID-19 pandemic, healthcare organizations implemented the Virtual Leadership (VL) approach, which involves using technology. As nurses have transitioned to remote work and relied heavily on technology for communication and collaboration, understanding how this shift has influenced their well-being and job satisfaction is paramount [12]. Virtual leadership has the potential to affect factors such as work–life balance, feelings of support and connectedness, and overall job engagement. It has become increasingly crucial to examine their impact on nurses’ psychological behavior during the COVID-19 crisis. Recently, the literature has highlighted the significant effects of the COVID-19 pandemic on nurses’ psychological and mental health [5]. Previously, the literature highlighted nurses’ psychological stress due to COVID-19 in tertiary hospitals in Pakistan. Therefore, it is essential to investigate the aspects that influence nurses’ psychological behavior during the COVID-19 crisis.

Due to health concerns and government restrictions, the COVID-19 crisis forced leaders to work from home. As a result, VL has become relevant for leading and managing teams remotely using technology for effective communication, collaboration, and decision-making while maintaining team cohesion and performance in delivering quality healthcare services [12]. In this context, virtual leadership refers to the positional leaders who make decisions that impact nurses’ overall operations and work behavior. However, the shift to VL has resulted in significant disruptions to the traditional leadership style. For instance, Fernandez et al. [13] noted that VL can contribute to a sense of isolation and disengagement among healthcare professionals. Hence, VL undoubtedly influences how nurses perceive their leaders’ guidance and decisions. VL involves healthcare executives, clinical managers, and nurse managers.

For instance, Laukka et al. [14] argued that VL had detrimental effects in healthcare sectors. They noted that nurses efficiently practiced traditional roles in patient care, coordination, advocacy, and teamwork during the traditional leadership style. During in-person leadership, nurses engage in regular face-to-face interactions with their leaders, including coaching and mentoring sessions, performance evaluations, and other forms of guidance and support that facilitate overall performance [13]. 

Similarly, Sandberg et al. [15] studied VL during the COVID-19 pandemic. Notably, they observed that the absence of face-to-face communication and appropriate feedback from leaders were prominent obstacles faced in VL settings. Hence, VL brings up new challenges in healthcare sectors concerning communication. If communication challenges are not properly managed, they may affect leadership functions such as influencing, decision-making, and supervision [16], consequently impacting the work behavior of nurses.

According to Grunes et al. [17], emotional intelligence (EI) is effective in adapting to and coping with stressful environments. An individual with strong EI can better resist psychological stress. A study by Mayer and Salovey [18] revealed that EI is a strong predictor of how ‘individuals’ deal with psychological stress. Researchers have conceptualized different EI models in the healthcare sector and confirmed their effective impact on controlling psychological stress among nurses [19,20]. Emotional competencies help minimize destructive consequences in hospitals, and the nurses’ overall job performance can be boosted substantially [21]. In that regard, the EI of nurses has been recognized as having a significant and influential effect on job stress levels, burnout levels, and job performance among nurses [13]. During emergencies, an emotionally intelligent individual knows how to react appropriately. Nurses face uncertain situations quite often. They need to display positive reactions, such as providing care to patients while managing their work stress and burnout level. The COVID-19 crisis posed many challenges to hospitals, enhanced psychological stress among nurses, and adversely affected job performance. Nevertheless, nurses’ EI traits can help manage unexpected changes and eliminate the influence of job stress and burnout level caused by VL practice.

A lot research has been conducted with regards to analyzing nursing performance during the COVID-19 pandemic in developed countries, specifically China, the UK, the USA, and European countries [9,22]. A few studies have examined psychological behavior among nurses in underdeveloped countries such as Pakistan [5]. However, to the best of our knowledge, the present study is the first to explore the impact of VL and EI on psychological stress among nurses in the health sector of Pakistan. As well as elaborating on the above discussion on work stress, burnout, and job performance, this study intends to discover the research gaps concerning virtual leadership. It also highlights that nurses’ emotional intelligence has a direct and moderating impact on the relationship between virtual leadership, work stress, burnout, and job performance.

The structure of this study is as follows: Section 2 provides a comprehensive review of the literature. Section 3 shows logical arguments and the development of our hypotheses along with the research model. Section 4 explains our research methodology, and Section 5 presents a statistical analysis and the results of our study. Section 6 contains a discussion of the results, draws conclusions, and describes the practical implications derived from our results as well as our study’s limitations, followed by future research directions.

## 2. Literature Review

### 2.1. Virtual Leadership

A leader is an individual who provides direction and guidelines and sets goals for subordinates or employees to accomplish organizational goals [23]. Similarly, the virtual leadership style refers to the management or leading of an organization with an intense emphasis on electronic communication rather than face-to-face communication [24]. Kandil and Moustafa [25] portrayed VL as a social-influence practice through advanced information technology to bring about changes in the psychological behavior of individuals. A VL approach begets counter-productive work behavior. Virtual leadership induces psychological stress, burnout, and mental exhaustion among nurses [26]. Several studies examined the influence of leadership style on employee effectiveness and productivity, such as innovativeness, commitment, job satisfaction, and motivation [27,28]. Virtual leadership affects employees’ physical and mental states, which results in work stress, burnout, and diminished job performance [29]. Because of the VL style, a disruption is created that affects employees’ work efficiency and innovativeness, making the firm or organization less efficient.

### 2.2. Emotional Intelligence

EI is an individual’s skill and ability to gauge their feelings and emotions, the emotions of others, and manage a favorable relationship with others [30]. Goleman [31] suggested that EI plays a fundamental role in the course of emotional labor. EI can adjust the behaviors and attitudes of employees in a workplace environment, reducing the levels of work stress and burnout and increasing overall job performance [32]. Emotions are crucial in nursing; nurses provide services in emotionally charged and challenging situations that can enhance their psychological stress. Scientific research considers the influence of EI among nurses. Dugué et al. [20] suggested that EI is a premium skill that enables nurses to cope with their job-related challenges. Previous studies, such as those by Codier et al. [33], measured EI in clinical practices and explored positive relationships between EI and job performance. Some studies revealed EI as a factor that reduced work burnout and improved well-being [32].

### 2.3. Work Stress

Work stress in nursing refers to the physical and emotional strain on nurses due to job demands, which can lead to negative effects on their health, job satisfaction, and productivity [34]. Job-related stress creates psychological issues. Compared with other occupations, nurses experience higher stress-related problems [35]. Laeeque et al. [2] showed that, when nurses experience psychological stress, absenteeism and turnover intention rise in healthcare sectors, negatively affecting their job performance [36].

### 2.4. Burnout

Gonçalves et al. [3] define Burnout as a state of emotional, physical, and psychological exhaustion among nurses resulting from stressful work situations that cannot be effectively managed. Barello et al. [9] stated that burnout consequences, for example, higher absenteeism and turnover, psychological problems, negative changes in overall work performance, or less job satisfaction [37], are widespread among nurses.

### 2.5. Job Performance

Job performance refers to how effectively and efficiently individuals carry out their duties and responsibilities in accordance with established standards [4]. Job performance is considered a significant parameter in the nursing profession. Nursing job performance refers to the ability of nurses to achieve work tasks, meet job expectations, deliver standard services, and accomplish organizational goals [4]. After conducting a vast review of the literature, it is evident that job performance is directly related to individual psychological behavior [21].

### 2.6. Theory of Emotional Intelligence (EI)

A previous study reported that employees displayed low intent and negative work behavior in a disruptive work environment compared to a cooperative work environment [38]. The healthcare environment is usually considered more disruptive. Nurses frequently experience high levels of occupational stress because of inadequate leadership, heavy workloads, conflicts, and insufficient supportive relationships [9]. Alam et al. [39] found that EI traits help to reduce the negative consequences in a work environment and significantly improve job performance among employees. Therefore, the present study’s hypotheses are based on the emotional intelligence model [18], which defines EI as a set of cognitive abilities used to perceive, practice, understand, and manage emotions in a certain situation. Individuals with strong EI abilities can effectively identify and control negative behavior. Some scholars identified high levels of psychological stress among nurses due to their challenging work environment. This indicates that such negative emotions affect their well-being and job performance. To encounter such challenges, nurses require theoretical knowledge, professional skills, and strong EI [40].

Mansel and Einion [41] suggested that EI can be used as a psychological tool to evade the complications brought about by specific leadership styles. A leadership style either positively or negatively impacts the work behavior of nurses. When employees perceive their leaders as ineffective in leadership style, it can create psychological stress that affects their overall job performance [42]. By leveraging EI skills, nurses can navigate the challenges of VL, promote team resilience, and ensure quality service despite the pressures of the virtual work environment [26]. Ayalew and Ayenew [43] further postulated that individuals with EI skills foster effective communication, teamwork, and conflict resolution in virtual work settings and that coping with these challenges can lead to better work performance. Furthermore, Foster et al. [44] suggested the relationship between the EI model and stressful factors and demonstrated that individuals with high-level EI have lower levels of perceived psychological stress.

A previous study has proved that EI skills and abilities can significantly diminish psychological stress and burnout in a challenging work environment [45]. However, scholars have not investigated how much the virtual leadership style affects nurses’ work stress, burnout, and job performance and how EI can influence their relationship among these variables.

## 3. Hypothesis Development

### 3.1. Virtual Leadership, Work Stress, Burnout, and Job Performance

Leadership style has been considered crucial in shaping organizational culture. Inadequate leadership can result in a lack of clarity and direction for employees, which in turn can lead to frustration and reduced motivation [46]. However, an appropriate leadership style provides accurate directions and supervision and sets objectives for the employees to achieve the organization’s goals [8]. COVID-19 has prompted many healthcare sectors to require their staff to work remotely, which has led to the switch from physical leadership to virtual leadership [47]. For instance, Falatah [10] points out that nurses received direct guidance in person during the traditional leadership style and that leaders were physically present in hospitals. However, the transition to virtual leadership poses challenges for employees. “For example, nurses rely on remote guidance through technology-mediated channels, which significantly impacts their daily work routine [15].” Kiljunen et al. [26] argued that during VL, individuals experience many deficiencies, such as inadequate communication, low trust, and poor coordination, compared with physical leadership. For individuals to overcome the difficulties of VL, employees in organizations need to collectively understand how to approach this unique leadership style.

Cortellazzo et al. [48] reported that the biggest challenge of VL is making employees perform collectively and creating a culture that allows all leadership opinions to be heard. However, Kiljunen et al. [26] indicated that some of the significant VL challenges in healthcare during the COVID-19 pandemic were communication, stress-management, burnout, and job performance. Cowan [29] suggested that nurses lack trust and confidence in virtual leadership, which causes psychological stress. Consequently, nurses experience more work stress and burnout, negatively affecting their job performance.

Liu et al. [34] indicated that VL was performed in developed nations before the spread of COVID-19. However, no healthcare sector was sufficiently organized for the extensive and sudden shift to virtual leadership. Existing evidence has proved that some organizational employees feel their skills have improved through VL compared to traditional leadership [49]. However, this response contrasts with the perspectives of nurses in Pakistan. Kiljunen et al. [26] suggested that virtual guiding or supervision arrangements are insufficient due to numerous challenges. Appropriate communication and technological facilities are not probable. Specifically, in rural regions where Internet services barely exist, VL is inadequate compared to physical practices [39]. Khan et al. [50] examined the psychological stress of Pakistani nurses with regards to leadership style. Their findings suggested that leadership style can adversely affect job performance in developing countries such as Pakistan.

The COVID-19 crisis affected the group learning activities between lab experiments and traditional healthcare meetings between leaders and nurses due to the shift to virtual leadership. This unexpected change led to nurses experiencing psychological stress and increased levels of burnout. Elyousfi et al. [51] discussed that VL enhanced occupational stress, significantly impacting nurses’ well-being, decision-making capabilities, emotional health, anxiety, depression, and workloads. Mansel and Einion [41] elaborated on the psychological impact of nurses in the healthcare sector during the COVID-19 crisis. They discovered that nurses faced huge psychological and emotional stress due to the high infection rate and deaths. Additionally, their psychological stress was exacerbated due to switching from the traditional leadership style to the VL style. Using a VL style in healthcare obviously changes the way nurses interact with leaders and other staff [52]. Virtual leadership provides a mutual framework that includes both technology and leadership. This framework was analyzed to explore the influence of virtual leadership on nurses’ psychological behavior during the COVID-19 crisis in Pakistan’s hospitals. Hence, the following hypotheses were proposed:

**Hypothesis** **1** **(H1).**
*Virtual leadership shows a positive effect on work stress among nurses.*


**Hypothesis** **2** **(H2).**
*Virtual leadership is positively linked to working burnout levels among nurses.*


**Hypothesis** **3** **(H3).**
*Virtual leadership negatively affects nurses’ job performance.*


### 3.2. EI, Work Stress, Work Burnout, and Job Performance

The nursing profession is emotionally demanding. Experimental research has discovered the potency of EI as a concept associated with favorable outcomes [53]. The concept of EI was highlighted in the mid-1920s, but it received more consideration once Salovey and Mayer [30] appropriately defined it by stating that EI is “the capability to observe one’s and others’ mental state and sentiments, distinguish among them, and practice these feelings to guide an individual’s thinking and reactions [54].” Nurses with high-level EI demonstrate greater self-awareness and the ability to reflect on their emotions, making them more resistant to stressful situations [55]. In addition, studies have highlighted that EI can moderate the effect of emotional needs on emotional exhaustion and cultivate self-efficacy in nursing contexts [56].

Preliminary investigations demonstrated that EI traits and aptitude play a favorable role in helping an individual adjust to a chaotic workplace. Bar-On [57] posited a few theories that display a link between emotions and an individual’s competency when dealing with stress and burnout in a hectic work environment. Previous research has discussed the significance of EI in the healthcare sector. The outcomes confirmed that a low level of EI was associated with higher levels of perceived stress and burnout among nurses [21]. A study by Rasool et al. [58] highlighted that a toxic work environment can result in employees experiencing depression, stress, burnout, and severe psychological issues, leading to absenteeism and poor organizational performance.

In hospitals, nurses often encounter emergencies; therefore, EI may help individuals perform quality services during emergencies. Contemporary healthcare organizations require nurses who possess strong EI and are able to manage workloads and deliver high-quality healthcare services [55]. Several studies have discovered that EI strongly correlates with high-stress management and well-being [40,59]. EI skills equip nurses with effective emotional management and interpersonal skills, contributing to positive work-related outcomes. Therefore, this study aims to explore the impact of EI on nurses’ work stress, burnout, and job performance during the COVID crisis. Hence, we hypothesized the following:

**Hypothesis** **4** **(H4).**
*There is a positive relationship between nurses’ emotional intelligence and a nurse’s job performance.*


**Hypothesis** **5** **(H5).**
*There is a negative relationship between nurses’ emotional intelligence and work stress.*


**Hypothesis** **6** **(H6).**
*There is a negative relationship between nurses’ emotional intelligence and burnout levels.*


### 3.3. Moderating Role of Emotional Intelligence

An emotional intelligence model is fundamental for organizational, psychological, and management studies. EI refers to a set of non-cognitive aptitudes and skills that reduce social and environmental burdens [60]. Due to COVID-19, hospitals were presented with incredibly stressful challenges. Hence, it is reasonable to suggest that emotional capabilities can help control the psychological stress experienced by nurses and boost their job performance [55]. Sadovyy et al. [59] revealed that employees with a high EI level accurately used their skills to cope with challenges related to the pandemic. However, past research is limited in determining the impact of nurses’ EI and leadership style on their psychological well-being, specifically in Pakistan. The concept of EI provides a theoretical framework for measuring the efficacy of a specific leadership style and nurses’ levels of work stress, burnout, and job performance. The significance of EI traits in nursing involves measuring how EI impacts work stress, burnout level, job satisfaction, and performance among nurses [41]. Due to the sudden implementation of virtual leadership practices, it is important to identify the impact of nurses’ EI on VL and their psychological well-being during the COVID-19 crisis; hence, the authors developed a research model, as shown in Figure 1.

A study by Wang et al. [61] explored the positive correlation between leadership style and nurses’ emotional intelligence. The research findings revealed that the interaction between EI in nursing and various leadership styles has a notable impact on work stress and burnout levels in the workplace, increases motivation among nurses, and encourages them to work towards achieving the organization’s goals [39,62]. Hurley et al. [19] claim that EI traits assist an employee in adjusting to different environments. Consequently, high-level EI diminishes the anxiety that frequently arises during stressful situations. Accordingly, regarding virtual leadership in healthcare sectors, a study by Sharpp et al. [52] discussed the negative influence of VL on nursing psychology. A lack of physical interaction between employees inhibits the creation of a strong working environment, which negatively influences work behavior among nurses. Previous studies have also demonstrated the impact of online platforms on individual psychological stress and well-being among students and teachers [63]. Unpredictable changes in work environments due to the pandemic negatively impacted the work behavior of individuals [64]; therefore, EI has a moderating effect on one’s ability to adapt to sudden changes in their workplace environment.

The University of Toledo [65] researched the pandemic. Their analysis confirmed that EI effectively moderates work stress and burnout level and contributes effectively to addressing the challenges of virtual leadership in an emergency. EI traits diminish the negative effects of virtual leadership on employee work behavior and mitigate adverse reactions of emotions, leading to better work performance [52]. Consequently, EI skills can help nurses navigate remote communication challenges and build strong relationships with their team members and virtual leaders. Ayalew and Ayenew [43] conducted a study on EI attributes (e.g., self-awareness, self-regulation, motivation, empathy, and social skills) and found that individuals with high levels of EI are less likely to be caught engaging in deviant behaviors than those with low-level EI. Thus, EI is considered a definite individual competency that moderates the impact of virtual leadership on nurses’ psychological issues in terms of work stress, burnout level, and work performance. Hence, we hypothesized the following:

**Hypothesis** **7** **(H7).**
*EI moderates the relationship between virtual leadership and work stress among nurses.*


**Hypothesis** **8** **(H8).**
*EI moderates the relationship between virtual leadership and burnout level among nurses.*


**Hypothesis** **9** **(H9).**
*EI moderates the relationship between virtual leadership and nurses’ job performance.*


## 4. Research Methodology

### 4.1. Research Approach

Using a convenience sampling technique, we employed a survey methodology by developing a questionnaire. We used this method because it is a popular and reliable research technique commonly used by scholars. This method is appropriate when obtaining a complete sampling frame is difficult. This type of sampling is suitable because it permits a theoretical generalization of the findings. The questionnaire survey method is cost-effective and suitable for contacting small and large groups, allowing the quick collection of data samples for statistical analysis [58].

### 4.2. Questionnaire Development

This research explores the impact of virtual leadership and emotional intelligence on work stress, burnout, and job performance among nurses in the tertiary hospitals of Pakistan. This research also examines the role of EI in moderating virtual leadership and nurses’ work stress, burnout, and job performance. A total of five latent variables and thirty-one items comprised a questionnaire that was adapted from previous studies, measured on a 5-point Likert scale ranging from 1 (strongly disagree) to 5 (strongly agree). Initially, we conducted a pilot study to check the reliability and validity of the questionnaire items. The participants reviewed the questionnaire items and suggested some amendments. The respondents’ recommendations were valid and considered, and the scale of the items was redeveloped and distributed for data collection.

### 4.3. Respondent Summary

The data sample for this research was drawn from nurses using a survey methodology. The participants worked in five tertiary hospitals in Punjab and Khyber Pakhtun Khwa, Pakistan. To better analyze the relationship structure between virtual leadership, emotional intelligence, work stress, burnout levels, and job performance, we gathered data from nurses in the clinical COVID-19 wards in each of the five Pakistani hospitals. The reasons for focusing on the healthcare sectors are as follows: (1) Nurses work in a stressful environment, and the COVID-19 crisis accelerated their levels of psychological stress, significantly decreasing their overall job performance [66]. (2) All of the healthcare employees experienced massive organizational changes due to the pandemic, such as shifting virtual leadership and virtual teams, which provided an opportunity to explore the relationship between the targeted constructs.

To strengthen the response level and avoid unbiased responses, some procedures were adopted (e.g., cover letter, confidential premises, privacy) to promise participants that their collected data would just be used for academic and research purposes. We followed a questionnaire methodology that involved self-reporting. In total, 370 questionnaires were provided to nurses who were willing to participate. Overall, 326 questionnaires were completed and returned, but only 274 were considered acceptable for further statistical analysis, considering a 99% confidence level, standard deviation value of 0.5, and ±1% margin of error. The remaining questionnaires were ignored because they did not fulfill the required criteria (e.g., error, missing data). Westland’s [67] statistical computing criteria recommends that the SEM model’s smallest value be 250 cases for the absolute sample size. Therefore, our data sample of 274 questionnaires meets the required minimum standard for sampling acceptability.

### 4.4. Demographics

Table 1 represents the demographic characteristics of the respondents. A total of 13% were male and 87% were female (35 and 239, respectively). Out of the 274 respondents, there were 46 nursing supervisors/senior nurses and 228 clinical nurses. All the participants were separated into three categories: college-level, undergraduate, and graduate, in terms of academic qualification. A total of 91 nurses had college-level degrees, which is 33.2%, 126 nurses were undergraduates (46%), while the remaining 20.8% were graduates. Their levels of job experience were also separated into four categories: those with less than three years experience accounted for 79 nurses (28.8%), those with 4–6 years experience accounted for 97 nurses (35.4%), those with 7–9 years accounted for 66 nurses (24.1%), and participants with more than ten years experience accounted for 32 nurses (11.7%).

### 4.5. Measurements

#### 4.5.1. Emotional Intelligence

We assessed emotional intelligence by using an eight-item scale outlined by Schutte et al. [68]. The items were carefully selected based on their reliability and validity to assess emotional intelligence. An example of a statement included in the questionnaire is “I truly understand my emotions”. Participants were asked to select their option for all statements correspondingly on a 5-point Likert scale ranging from 1 (“strongly disagree”) to 5 (“strongly agree”). The overall Cronbach’s alpha reliability was 0.758.

#### 4.5.2. Virtual Leadership

Virtual leadership was measured with a seven-item scale adapted from Lovelace [69] and He [70]. The item scale was modified to suit our study context. Another example statement is, “VL communication helps me to engage in my task.” Nurses were asked to answer questions about how they understood each virtual leadership practice. The overall Cronbach’s alpha reliability was 0.752.

#### 4.5.3. Work Stress

To consider this measurement, Lazarus and Folkman’s [71] five-item scale of work stress was used to gauge nurses’ work-related stress, physical health, and well-being. Additionally, Nurses were queried to specify how often they had fears and feelings of COVID-19 symptoms (e.g., headache, coughing, fever). The scale was constantly considered from “1”, though not for all questions, to “5”. The Cronbach’s alpha was 0.793.

#### 4.5.4. Burnout

Schaufeli et al. [72] developed a five-item scale for work burnout. A sample statement from the questionnaire is as follows: “During work, my energy level is deficient, and feeling exhausted.” The Cronbach’s alpha of this scale was 0.791.

#### 4.5.5. Job Performance

To evaluate the work performance of nurses, six-item scales determining work engagement, physical and emotional satisfaction, self-control, and task achievements were adopted from a study by Singh et al. [73]. The queries were constructed based on the COVID-19 crisis and included statements such as “I feel satisfied with the current supervision I have in my job”. A Likert-type scale ranging from 1 (poor performance) to 5 (excellent performance) was used to measure participant responses. The overall value of Cronbach’s alpha was 0.749.

## 5. Results

### 5.1. Testing Measurement Model

For statistical analysis, SmartPLS 3.3.9 was used in this research to analyze the relationship between the targeted variables. SmartPLS 3.3.9 is statistically more effective and relatively less sensitive to sample sizes than other software used for variance-based analyses [57]. We tested and confirmed the modeling procedure involved in two practices: (1) the measurement model test and (2) the structure model test by running partial least square structural equation modeling (PLS-SEM). Initially, we examined the measurement model to analyze its convergent validity [74]. All items were measured by scale factor loadings, Average Variance Extracted (AVE), and Composite Reliability (CR). The factor loading of all items ranged from 0.681 to 0.855, confirming the minimum criteria and satisfying the standard value of 0.6 outlined by Chin et al. [75]. CR specifies the extent to which the indicators converge to identify the internal consistency, ranging from 0.827 to 0.892, meeting the least standard criteria of 0.7. AVE, which indicates the variance explained for the latent construct, was in the range of 0.576 to 0.654, exceeding the smallest range of 0.5 as recommended by Hair et al. [76] and shown in Table 2.

Since the data were collected from the same respondents for both predictor variables and dependent variables through the same instrument/questionnaire using the same method, there may be a problem in the form of common method bias [77]. The Harman single-factor test was performed with SPSS 16.0 to check if there was an overall method bias. Harman’s single-factor test is a non-rotated exploratory factor analysis performed on an instrument or questionnaire. In this test, if a single factor cannot explain most (0.50) of the variance, there is no general bias problem. In this study, the single factor could only explain 39% of the variance of the whole instrument; therefore, there is no common method bias. A Square Root Mean Residual (SRMR) value of less than 0.09 is considered to fit the model well [78]. This model fitted well, with a SRMR score of 0.069 for the saturated model and 0.074 for the estimated model.

Moreover, discriminant validity was also analyzed, which provided a low correlation among the targeted variables. Various criticisms have indicated that Fornell and Larcker’s [79] standard criteria do not reliably determine the inadequacy of discriminant validity in ordinary research. Therefore, an alternate technique to determine the discriminant validity was recommended by Henseler et al. [80]; the heterotrait-monotrait ratio of correlations was examined for discriminant validity. The HTMT value was within the standard criteria of lesser than 0.85 [81], as shown in Table 3. Consequently, each variable confirmed the reliability and indicated a reliable measurement model.

### 5.2. Assessing Overall Model Fit

The structural model analysis examined the relationship among the considerate variables. To explore the relationship between the dependent, independent, and moderating variables, a PLS-SEM structural model was run [74]. R^2^, β, and *t*-values were observed through the 5000 bootstrapping methods [82]. Initially, we perceived the interactions among the variables. As shown in Table 4, the fallouts of Hypothesis 1 and 2 elucidated the exerting effect of virtual leadership on work stress (β = 0.36 *p* < 0.00) and burnout level (β = 0.31, *p* < 0.01). In contrast, Hypothesis 3 shows the negative influence of virtual leadership on job performance (β = −0.29, *p* < 0.01). Furthermore, the impact of EI on job performance (β = 0.41, *p* < 0.01) indicates a significant positive relationship; however, the influence of EI on work stress (β = −0.33, *p* < 0.00) and burnout (β = −0.38, *p* < 0.05) significantly revealed a negative relationship. Therefore, our results validate Hypotheses 4, 5 and 6, respectively.

Furthermore, virtual leadership explains 35.3% of the variance in work stress (R^2^ = 0.353). In contrast, virtual leadership explains 40.1% of the variance in work burnout (R^2^ = 0.401) and 42.5% in job performance (R^2^ = 0.425), as shown in Figure 2. The R^2^ values of 0.353, 0.401, and 0.425 exceed the recommended range of 0.26 [83], which indicates a considerable model. Next, we evaluated the effect sizes (f^2^) (see Table 4); the *p*-value suggests significant interactions but it does not show the size of an effect. Hence, scholars have had disputes with the data interpretation and results. It is recommended that changes in R^2^ must be observed [76]. Cohen [83] suggested a criteria for analyzing the effect size and specified 0.02 for small effects, 0.15 for medium effects, and 0.35 for large effects. Q^2^ shows how the data can be organized using the model and the PLS constraints. A Q^2^ value higher than zero indicates the predictive significance of the model, whereas a Q^2^ value lower than zero suggests a lack of predictive significance in the model, as given in Figure 2. A Q^2^ for all three variables shows satisfactory predictive relevance.

Finally, the present research hypothesized that EI moderates the relationships between virtual leadership, work stress, work burnout, and job performance among nurses. To analyze the appropriate analysis, the PLS product-indicator method was run for hypothesis assessment; PLS can provide an accurate assessment for moderation impact [84]. To check the moderating effect, virtual leadership, and EI were multiplied to create an interaction construct that predicts work stress, work burnout, and job performance. The anticipated standardized path coefficients for the impact of the moderator on the work stress (β = −0.21; *p* < 0.01), work burnout (β = −0.23; *p* < 0.05), and job performance (β = 0.26; *p* < 0.05) were significant, as shown in Table 4. Emotional intelligence moderates the relationships between virtual leadership, work stress, work burnout, and job performance among nurses. Consequently, H7, H8, and H9 were also validated.

## 6. Discussion

Previous research on leadership style has explored the major influences of psychological behavior. We have explained the impact of the virtual leadership style on work stress, burnout, and job performance among nurses. Virtual leadership is an inadequate leadership style that affects psychological work behavior among nurses. Moreover, this research also shows EI’s direct and moderating impact on the relationship between virtual leadership, work stress, burnout, and job performance.

First, we measured the direct impact of virtual leadership on nurses’ work behaviors. The results revealed that virtual leadership positively impacts work stress and burnout and negatively impacts nurses’ job performance, supporting H1, H2, and H3. Our findings indicated that sudden changes in leadership style within organizations strongly influences the psychological stress of nurses, which affects their work performance and well-being, specifically during times when hospitals are under immense strain, such as the pandemic. Our findings support the previous literature. Rad and Yarmohammadian [85] showed that leadership style is among the most critical factors in healthcare sectors. An Inadequate leadership style can increase costs, reduce effectiveness, and create dissatisfaction among nurses.

Similarly, Alam et al. [39] conducted a study on the influence of the virtual environment on individual work behavior. Their study showed that a virtual work environment negatively affects individual psychological behavior and work performance. However, to control psychological stress and burnout, nurses need to manage their negative emotions and show a positive work intention, which helps them increase their work efficiency [59]. Sharpp et al. [52] demonstrated that a virtual work practice creates trouble in the healthcare organization, and the employees tend to quit rather than improve their work performance. Therefore, the above discussion indicates that virtual leadership enhances work stress and burnout and negatively affects nurses’ job performance, supporting our findings.

Second, this research confirmed that EI significantly influences nurses’ work behavior. Hence, the findings of our study support H4, H5, and H6 and are in line with the previous research that has found a significant impact of individual EI on work-related outcomes [68,86]. A recent study by Ahuja et al. [87] also supports our study argument that VL can make individuals feel isolated and disconnected, if not appropriately managed, which can consequently enhance stress and impact the well-being of the workforce in the organization. Nurses with high EI can regulate their emotions based on the needs of the situation, which helps keep them involved in work and keeps them physically and mentally strong. Based on the findings of this research, as well as those found in the literature [88], it can be said that emotionally intelligent nurses are more well-equipped to fulfill the needs and demands of patients and take a proactive approach when responding to emergencies.

Moreover, we examined the moderating role of EI between virtual leadership and work stress, burnout, and job performance. Our study confirmed the moderating role of nurses’ EI in virtual leadership, work stress, burnout level, and job performance, consequently supporting H7, H8, and H9. Our result revealed that nurses with low EI experienced higher levels of work stress and burnout than those with higher emotional intelligence. EI can develop psychological competencies that help individuals control their work behavior in a challenging work environment. This research also confirms that nurses with high-level EI are less impacted by the negative effects of virtual leadership. This is consistent with previous observations. Alam et al. [39] claim that EI regulates emotions appropriately during unexpected situations, such as the COVID-19 pandemic. EI could assist nurses in coping with the psychological stress induced by the virtual leadership style [88]. Our findings are parallel to those of Barello et al. [9]. They argued that emotionally intelligent individuals could appropriately adapt to different leadership styles. Sun et al. [89] confirmed that during the pandemic, nurses experienced significant psychological stress, but emotional intelligence (EI) can buffer the negative effects of stress and burnout. Furthermore, Soto-Rubio et al. [88] suggested that EI traits are significant in any profession, but most importantly in healthcare sectors, where EI is negatively correlated to job stress, reduced risk of work burnout, and influences job performance among nurses.

### 6.1. Conclusions

To conclude, this research has shown the relationships between virtual leadership, emotional intelligence, work stress, work burnout, and job performance of nurses employed in the COVID-19 ward. Our findings demonstrate that virtual leadership strongly influences nurses’ work stress and burnout while having a negative impact on job performance during the COVID-19 pandemic. Our results also revealed that emotional intelligence enhances job performance and eases nurses’ psychological hassle and burnout levels.

Based on the significant outcomes, the results suggest virtual leadership has a detrimental effect on nurses’ work behavior in the healthcare sector. The lack of face-to-face interaction and support, communication issues, limited social interaction, and team cohesion can all lead to nurses experiencing difficulties with achieving standard job performance in a virtual work environment. These factors highlight the challenges and limitations of virtual leadership, indicating the need for additional strategies to address the negative impact of it and promote effective leadership in virtual settings. Moreover, our findings indicated that EI is an important moderator that influences nurses’ work behavior. This highlights the importance of considering EI as a factor that can shape nurses’ psychological empowerment in healthcare settings. 

Consequently, our study confirmed the accuracy of EI theory by showing that it strengthened the ability of individuals to prevent psychological stress and put their EI skills to use during emergencies. Therefore, EI training and development programs should be considered in the curriculum of the nursing profession. These programs can provide nurses with the necessary skills to recognize, understand, and manage their emotions effectively and navigate complex interpersonal dynamics in the healthcare setting. This can boost nurses’ emotional skills and help them resist the crucial effects (work stress, turnout intention, and burnout level) linked with the high extent of psychological problems experienced by nurses during virtual leadership. By possessing high emotional intelligence (EI) skills, nurses are better equipped to handle the psychological challenges associated with their work. These elements are useful for nurses’ physical and psychological well-being and allow nurses to effectively cope with emergency situations, improving their job performance.

### 6.2. Contributions, Limitations, and Future Research

#### 6.2.1. Theoretical Contribution

Our research findings help to understand that leadership style and nurses’ emotional intelligence are crucial predictors of their work behavior. The evidence suggests that the virtual leadership style is a major psychological problem for nurses in the healthcare sector [25], and nurses’ ability to control his/ her emotion is an important factor capable of reducing work stress and burnout [41,90]. The evidence suggests that EI traits could enhance nurses’ job performance in the workplace [20]. 

In addition, based on Mayer and Salovey’s [18] emotional intelligence theory, this study explores how to cope with the negative impacts of virtual leadership. Previous research has investigated the direct relationship between leadership styles, job satisfaction, and employee work engagement. However, the impact of the virtual leadership style in healthcare has been explored for the first time. Specifically, this study considered EI as a moderating construct between virtual leadership and nurses’ work behavior. 

Based on the empirical results, virtual leadership, and nurses’ emotional intelligence significantly influenced their work behavior during the COVID-19 pandemic. These results provide strategic directions for nurses in dealing with psychological stress caused by an inadequate leadership style and further confirm the practical value of emotional intelligence theory.

#### 6.2.2. Practical Implications

This research has some practical implications that can be derived from our results. This study is significant for leaders and healthcare professionals as it may help them to understand what influences nurses’ psychological work behavior. Those nurses with high emotional intelligence display a more positive work attitude under the influence of virtual leadership than the nurses with low EI skills. Accordingly, healthcare professionals should understand the negative impact of the virtual leadership style in healthcare organizations. Hospital management must formulate a program with an advanced leadership style that can credibly and adequately integrate with nursing, such as hiring nurses with strong awareness and EI capabilities. Enhancing and developing those characteristics that keep nurses from extensive psychological problems seems essential. Controlling psychological stress is important for nurses’ physical and mental well-being, which can engender innovative work behavior. During training and workshops, components of EI theory (self-awareness, self-regulation, motivation, empathy, and social skills) can build trust and confidence among employees, encouraging them to help each other during crises.

#### 6.2.3. Limitations and Future Research

The present study has some limitations. First, a notable limitation of this study involves the sampling technique. We employed a convenience sampling technique, and the participants were nurses currently employed in COVID-19 wards. Based on the self-reported analysis, the possibility of biased responses needs to be considered when interpreting the answers to the survey. Secondly, the results of this research only examined five government hospitals. In future studies, many private hospitals or other corporations or manufacturing sectors can be considered. The targeted population in this research model includes only nurse professionals. This model could be used in future studies in other professions, such as industrial employees, banking sector professionals, or other multinational companies. Third, we depended on survey data collected from hospitals in Pakistan. The authors may explore similar research in different regions, such as African and Western regions, in the future.

Moreover, this study examined EI as a moderator between virtual leadership and the psychological work behavior of nurses. Thus, it would be more interesting to determine the influence of other variables, such as cultural intelligence and organizational conflict. This research only investigated psychological work behavior from the perspective of nurses and did not involve top-level management. The last notable limitation of this research is the small sample data. We collected and analyzed the sample size from only five hospitals in Pakistan, and the sample size comprised 274 nurses, which could not be considered a large amount of data. There will be a larger sample size in future studies, and data will be collected from several hospitals.

## Figures and Tables

**Figure 1 healthcare-11-01537-f001:**
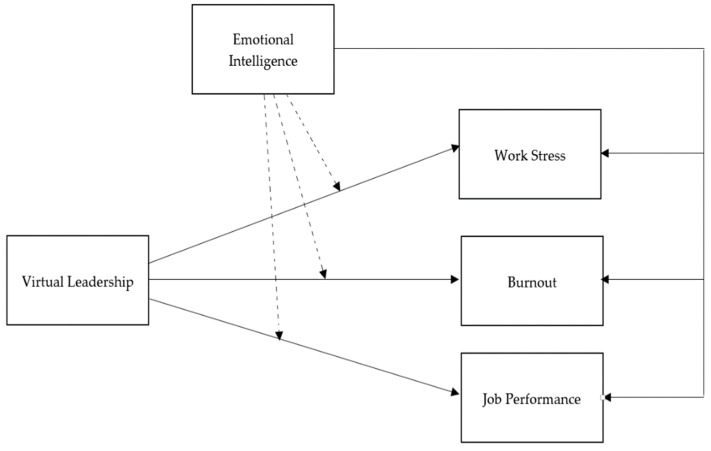
Theoretical framework.

**Figure 2 healthcare-11-01537-f002:**
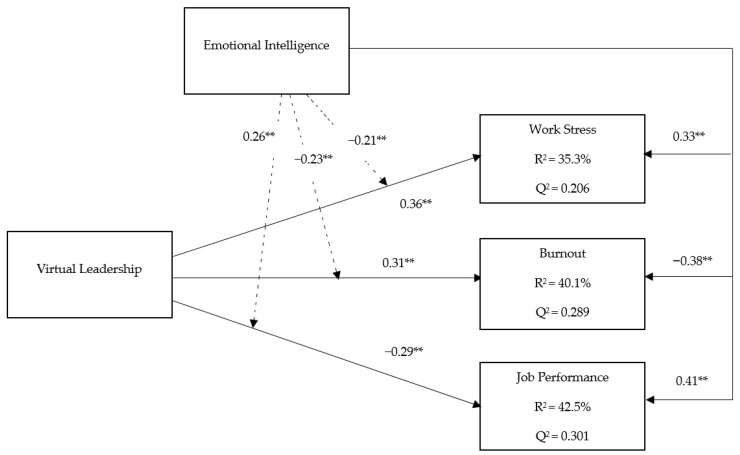
Structural model. Notes: ** *p* < 0.01.

**Table 1 healthcare-11-01537-t001:** ‘Participants’ demographic characteristics (*n* = 274).

Characteristics	Categories	*n* = 274	Percentage (%)
Gender	Male	35	13
	Female	239	87
Academic Qualification	College-level	91	33.2
	Undergraduate	126	46
	Graduate	57	20.8
Organizational Position	Supervisor/Senior Nurse	46	16.8
	Clinical Nurse	228	83.2
Job Experience	Less than 3 years	79	28.8
	4–6 years	97	35.4
	7–9 years	66	24.1
	Above 10 years	32	11.7

**Table 2 healthcare-11-01537-t002:** Reliability and validity.

Constructs	Items	Loading	Alpha	AVE	CR
**Emotional intelligence**	EI1	0.741	0.758	0.576	0.831
	EI2	0.752			
	EI3	0.749			
	EI4	0.763			
	EI5	0.795			
	EI6	0.771			
	EI7	0.739			
	EI8	0.761			
**Virtual leadership**	VL1	0.757	0.752	0.609	0.827
	VL2	0.804			
	VL3	0.826			
	VL4	0.813			
	VL5	0.796			
	VL6	0.733			
	VL7	0.729			
**Work stress**	WS1	0.791	0.793	0.653	0.892
	WS2	0.853			
	WS3	0.855			
	WS4	0.842			
	WS5	0.698			
**Burnout**	BO1	0.829	0.791	0.654	0.873
	BO2	0.830			
	BO3	0.816			
	BO4	0.754			
	BO5	0.813			
**Job performance**	JP1	0.681	0.749	0.600	0.839
	JP2	0.824			
	JP3	0.816			
	JP4	0.831			
	JP5	0.736			
	JP6	0.751			

**Table 3 healthcare-11-01537-t003:** Means, Standard Deviations, and Correlations *n* = 274.

	Variables	Mean	SD	1	2	3	4	5
**1**	**Job Performance**	3.47	0.43	1				
**2**	**Burnout**	3.01	0.49	−0.23	1			
**3**	**Work Stress**	3.16	0.37	−0.21	0.28	1		
**4**	**EI**	4.22	0.31	0.36	−0.43	−0.38	1	
**5**	**VL**	4.17	0.79	−0.39	0.38 **	0.29	−0.23 *	1

Notes: * *p* < 0.05; ** *p* < 0.01

**Table 4 healthcare-11-01537-t004:** Hypothetical Relationships.

Hypothesis	Beta	T-Value	*p*-Value	F Square	Decision
VL -> Work stress	0.36	2.513 **	0.00	0.071	Approved
VL -> Burnout	0.31	3.151 **	0.01	0.131	Approved
VL -> Job performance	−0.29	2.895 **	0.01	0.114	Approved
EI -> Work stress	−0.33	4.293 **	0.01	0.174	Approved
EI -> Burn out	−0.38	3.125 **	0.01	0.125	Approved
EI -> Job performance	0.41	4.591 **	0.05	0.197	Approved
VL * EI -> Work stress	−0.21	2.324 **	0.01	0.052	Approved
VL * EI -> Burn out	−0.23	2.697 **	0.05	0.089	Approved
VL * EI -> Job performance	0.26	3.926 **	0.05	0.153	Approved

Notes: * *p* < 0.05, ** *p* < 0.01.

## Data Availability

The datasets analyzed during the current study are available from the corresponding author on reasonable request.

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
