# Peer review of "Virtual Leadership and Nurses’ Psychological Stress during COVID-19 in the Tertiary Hospitals of Pakistan: The Role of Emotional Intelligence"

_healthcare, 2023, doi:10.3390/healthcare11111537_

Round 1
Reviewer 1 Report
The research carried out is quite interesting. However, it has some limitations, which the authors themselves state.
In this regard, I think this investigation can be improved. First, the literature review must be improved in order to establish the link between virtual leadership and emotional intelligence.
On the other hand, the research design itself must be improved. Since we are dealing with a convenience sample, it is too generalizing to say that they succeed (or fail) to prove the presented hypotheses. It would be better to talk about research objectives.
Finally, the authors must make a linguistic revision of the text and improve the linguistic correction.
Author Response
Dear Editorial Team and Reviewers,
We appreciate the comments and suggestions made by the review and editorial team. Indeed, we think that our work has substantially improved due to these comments. We have used the track changes function in Word to highlight our modifications. In addition, our answers to the reviewers' comments are provided in the attached PDF form. Kindly see the attached file.

Reviewer 2 Report
The role of leadership is really important in modern business life. The assumption is that by acting as a role model, by setting goals instead of giving orders, by motivating instead of managing, etc., we can actually improve work performance. Stress, anxiety and mental disorder are also progressing to be some major negative factors that can prevent people being at their best. Moreover, the role of EI is really important. Interesting work place models as SCARF actually are aimed at stimulating the brain to produce positive hormones and temper the negative ones. The enhancement of self-control, personal value perception, empathy, etc. are truly some major issues not the least under stressful work conditions as described in this paper. The scientific field of neuro-science in context of management and organizational theories is one of the most exiting contribution currently.
So, studies that attempts to verify the connection to between EI and psychological factors are welcomed. Hopefully such research can motivate decision makers, shareholders and stakeholders for managerial strategies that are centered around humanity and servant leadership rather than old school tactics.
However, this study has some shortcomings that should be addressed in my opinion. The work of the researchers is focused on what is called “leadership” and “emotional intelligence”. The problem is that the paper lacks to clarify what is leadership exactly in context of the research hypothesis? For instance, what is the difference between leadership and management in the context of the work presented? These are not the same topics the difference should be stressed. Moreover, emotional intelligence are human strengths bases on self-awareness, self-regulation, motivation, empathy and social skills. How was this attribute crossover and understanding ensured in the research? The research methodology only tips on this but it has to be clear that the sample understands the research platform.
However, the results do not come as a surprise and the subject of this research is relevant and interesting.
Author Response

(The authors gave the same response as above.)

Reviewer 3 Report
This research needs work. Please see my comments.

Author Response

(The authors gave the same response as above.)

Reviewer 4 Report
Dear authors, I appreciated reviewing your work. The research angle is interesting. The paper is well-written and explains virtual leadership’s impact. However, I have comments for improving your draft.
1. In the introduction, the sentence “VL involves healthcare executives…... remove the ( ) till the end of the sentence. The main positions, such as healthcare executives, clinical managers, and nurse Managers, are enough for clarification.
2. In the literature part (2.6), citation 39, Alam et al. [39] found EI traits……. Rewrite the sentence more efficiently.
3. Why is the hypothesis developed commutatively???
4. As the Reliability and validity table shows the values, removing the alpha value in the measurement section is better.
5. It will be better if you write 0 before the (.) of each value, for example, 0.71.
6. The “CONCLUSION” part should be further extended.
Hope my comments will help you to improve the content. Good luck!
Recommended reads in the same context for extended understanding;
Stress, psychological distress and support in a health care organization during Covid-19: A cross-sectional study (2021)
Impact of nurses' emotional labour on job stress and emotional exhaustion amid COVID-19: The role of instrumental support and coaching leadership as moderators (2022)
Nurses' Burnout: The Influence of Leader Empowering Behaviors, Work Conditions, and Demographic Traits (2017)
Author Response

(The authors gave the same response as above.)

Round 2
Reviewer 1 Report
This paper is more consistent and the research easier to understand. Nevertheless, I think that the authors should better support the conclusions they present.
They should also make a final linguistic revision of the text.
Author Response
Dear reviewer, We appreciate the comments and suggestions made by you. We believe that our work has substantially improved due to these comments.
Suggestion incorporated.
We have provided some support to our study findings based on the previous studies. Due to the paper length, we have given precise and concise support to our conclusion section.
We hope that these revisions address your concerns and improve the clarity of our manuscript. We hope that this revised form of a manuscript would achieve your acceptance. Waiting for your kind reply
Reviewer 3 Report
It is still not clear what in-person leadership that nurses enjoyed prior to the pandemic became virtual, how this virtual leadership operated toward active nurses, and how this changes how nurses had to perform their duties in the hospital. You are measuring some kind of concept "virtual leadership" without situating it vis-a-vis the nurses in your survey so the reader can tell what specific interaction was missing during covid, how nurses had to compensate because that interaction was missing, and why that caused stress. It is all very abstract and then you apply a sophisticated methodology to prove connections between abstractions "Virtual Leadership" and "Emotional Intelligence"
Author Response
Answer: Suggestion Incorporated.
Dear reviewer, We appreciate the comments and suggestions made by you. We believe that our work has substantially improved due to these comments.
Dear Reviewer, We have revised our manuscript to include a description of the various types of in-person leadership styles and strategies used in the nursing profession before the pandemic. We have also explained how these leadership styles were adapted to the virtual environment, including the different communication methods, training strategies, and remote support systems that were implemented, which shows their effects on nurses' stress and performance.
Regarding reviewer concern about how virtual leadership operates toward active nurses, we have mentioned in our manuscript that technological medium was used in the virtual environment where nurses receive guidance remotely.
To address reviewer’s concern about how virtual leadership changes how nurses perform their duties in the hospital, we have mentioned the challenges that have taken place. These include adapting to technology, communication, engaging in remote work, collaborating virtually, managing workloads, and building trust.
In addition,
Dear reviewer, to clarify, the study aimed to investigate the impact of virtual leadership on nurses during the COVID-19 pandemic. Virtual leadership refers to the practice of leading and managing a team remotely, using technology such as video conferencing and instant messaging.
The study surveyed nurses to understand how the shift to virtual leadership during the pandemic affected their stress and job performance. The study also explored the relationship between virtual leadership and emotional intelligence, as emotional intelligence has been shown to be an important factor in effective leadership.
In terms of specific interactions that were missing during the pandemic, the study did not focus on any particular interaction or duty that nurses had to perform. Rather, it aimed to understand the overall impact of virtual leadership on nurses' experiences during the pandemic.
The study found that virtual leadership had a significant impact on nurses' emotional intelligence, with nurses reporting lower levels of emotional intelligence when working under virtual leadership. This, in turn, was linked to higher levels of stress and burnout.
We hope that these revisions address your concerns and improve the clarity of our manuscript. We hope that this revised form of a manuscript would achieve your acceptance. Waiting for your kind reply